# LHC searches motivated by recent $B$-anomalies

**Darius Alexander Faroughy[1]⋆**

**1** J. Stefan Institute, Jamova cesta 39, Ljubljana, Slovenia

⋆ darius.faroughy@ijs.si

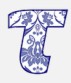 *Proceedings for the 15th International Workshop on Tau Lepton Physics,*
## Abstract

We discuss the physics case for the LHC motivated by the $B$-physics anomalies. After correlating semi-tauonic $B$ decays to di-tau production at the LHC, and discussing the possible models solving the $B$-anomalies, we show how existing LHC data in $\tau\bar{\tau}$ tails exclude most beyond the SM scenarios except for a handful of leptoquark (LQ) models. We analyze the impact of LHC searches for some of these LQ solutions using current data. In particular, we focus on the well known $U_1$ vector LQ as well as the GUT-inspired scalar LQs, $R_2$ and $S_3$. By exploiting the complementarity between di-tau searches and the lepton flavor violating decays $B \to K\mu\tau$ and $\tau \to \mu\phi$ we argue that these model can be cornered by the LHC, Belle II and LHCb in the near future.

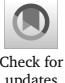
---

## 1   Introduction

In recent years we have noticed a growing interest in model building for lepton flavor universality (LFU) violation. This interest has been stimulated by a series of striking hints of LFU violation in a number of different experiments in the semi-leptonic decay channels of the $B$-meson. Experiments by BaBar [1, 2], Belle and LHCb [3–7], in which they measured the LFU ratio

$$R_{D^{(*)}} \equiv \frac{\text{Br}(B \to D^{(*)} \tau \bar{\nu})}{\text{Br}(B \to D^{(*)} l \bar{\nu})}\bigg|_{l \in \{e, \mu\}}, \tag{1}$$

indicate a combined excess in the tree-level process $B \to D^{(*)} \tau \bar{\nu}$ of approximately $3.8\,\sigma$ with respect to the SM values. Another indication of LFU violation has been reported for the flavor changing neutral current (FCNC) process $b \to sl\bar{l}$. LHCb [8, 9] measured $R_{K^{(*)}} = \text{Br}(B \to K^{(*)} \mu\mu)/\text{Br}(B \to K^{(*)} ee)$ reporting a $\approx 2.5\,\sigma$ deficit with respect to the SM prediction. If in upcoming experiments these departures from LFU are confirmed in $b \to c\ell\bar{\nu}$ and/or $b \to s\ell\ell$ transitions, this would clearly indicate the presence of physics beyond the SM.

Of many recent attempts by the theoretical community to provide a combined explanation of the $B$-anomalies, only a handful of models turn out to be viable. Part of the difficulty arises because New Physics (NP) solving the $R_{D^{(*)}}$ anomaly point towards new particles with masses below a few TeV, while the $R_{K^{(*)}}$ anomaly point towards a much heavier NP scale, up to an order of magnitude higher. One consequence of this dichotomy is that only the charge current anomaly has a very solid physics case for direct searches at the LHC. For this reason I will focus here exclusively on LHC searches relevant for the $R_{D^{(*)}}$ anomalies, however while keeping in mind models that provide a combined explanation of both anomalies. Using effective field theory and simplified models we demonstrate the usefulness of LHC searches in di-tau tails for testing different solutions to the $R_{D^{(*)}}$ anomaly. In fact, current LHC limits single out leptoquark (LQ) solutions as the most viable candidate. We also discuss how future di-tau searches at the HL-LHC combined with low energy searches for lepton flavor violating (LFV) $B$ and $\tau$ decays at Belle II and LHCb can ultimately test some of these LQ models in the near future. This proceedings is mainly based on the high-$p_T$ phenomenology in Ref. [10–12].

## 2   Effective theory

### 2.1   Low-energy effective theory

The leading non-renormalizable interactions describing semi-leptonic decays $d_i \to u_j$ below the electro-weak scale is given by the low-energy effective Hamiltonian

$$\mathcal{H}_{\text{eff}}^{d_i \to u_j \ell \nu} = -2\sqrt{2}\, G_F V_{ij} \Big[ (1 + g_{V_L}) \mathcal{O}_{V_L} + g_{V_R} \mathcal{O}_{S_L} + g_{S_L} \mathcal{O}_{S_R} + g_{S_R} \mathcal{O}_{S_R} + g_T \mathcal{O}_T \Big] + \text{h.c.}, \tag{2}$$

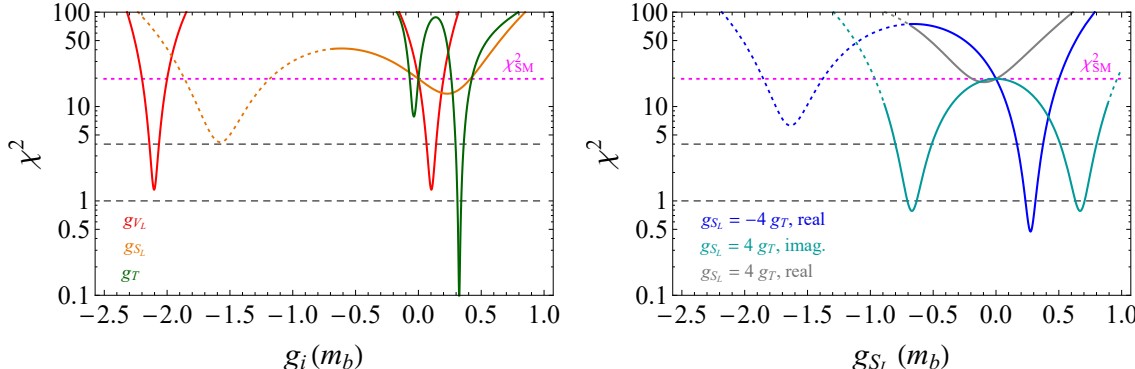

Figure 1: $\chi^2$ values for each individual effective coefficients fits to $R_D$ and $R_{D^*}$, compared to the SM value (magenta dotted line). In the left panel, $\chi^2$ is plotted against $g_{V_L}$, $g_{S_L}$ and $g_T$ at $\mu = m_b$. In the right panel, $\chi^2$ is plotted against $g_{S_L}(m_b)$ by assuming $g_{S_L} = \pm 4\, g_T$ at $\mu = 1$ TeV, for purely imaginary and real couplings. The dashed portions of the curves correspond to the values excluded by the $B_c$-lifetime constraints.

where $V$ is the CKM matrix and

$$\mathcal{O}_{V_X} = (\bar{u}\,\gamma^\mu P_X\, d)(\bar{\ell}_L \gamma_\mu \nu_L), \tag{3}$$

$$\mathcal{O}_{S_X} = (\bar{u}\, P_X\, d)(\bar{\ell}_R\, \nu_L), \tag{4}$$

$$\mathcal{O}_T = (\bar{u}_R\, \sigma^{\mu\nu} d_L)(\bar{\ell}_R\, \sigma_{\mu\nu} \nu_L), \tag{5}$$

are the four-fermion vector, scalar and tensor operators, respectively, $P_X$ with $X = \{L, R\}$ are the fermionic chiral projector to the left-handed (LH) and right-handed (RH) field components, $u$, $d$ and $\ell$ are the generic up-quark, down-quark and charged leptons fields for which we have omitted the flavor indices and $g_I$ with $I \in \{V_L, S_X, T\}$ are the Wilson coefficients.

We now specialize to the semi-tauonic $b \to c$ transitions and fit the coefficients $g_{V_X, S_X, T}$ to $R_{D^{(*)}}$ assuming negligible NP contributions to $b \to c\,(e, \mu)\bar{\nu}$ decays. In Fig. 1 (left), we show results of the one-parameter fits to each Wilson coefficient at the scale $m_b$. The dashed portion of the curves correspond to the exclusion limits from the $B_c$ lifetime on the branching ratio of $\mathrm{Br}(B_c \to \tau\,\nu) < 30\%$ [13, 14]. The only single operator that can explain the charged current anomaly is $\mathcal{O}_{V_L}$ (red curve) with $V - A$ structure. The tensor operator $\mathcal{O}_T$ can also fit the anomaly, but nonetheless is always generated in combination with scalar operators after integrating out the heavy NP state (see for example [15]). The anomaly can be successfully accommodated by scalar and tensor operators with Wilson coefficients satisfying $g_{S_L} = \pm 4 g_T$ at the NP scale (taken her at $\mu = 1$ TeV). As shown in Fig. 1 (right) [11], after running from the TeV scale down to $m_b$ we obtain one real solution (blue curve) and one imaginary solution (cyan curve) that fit very well $R_{D^{(*)}}$. Here, and through out this paper, we omit operators with light RH neutrinos. For these type of models the reader is referred to [16–19].

## 2.2 SM effective theory

In order to explore NP above the electro-weak breaking scale, it is necessary to restore the full SM gauge symmetry and work with the SM effective field theory (SMEFT) framework. In the Warsaw basis [20], the complete set of dimension-6 operators giving rise to semi-leptonic

$d_i \to u_j$ transitions are given by:

$$
\begin{aligned}
\mathcal{L}_{\mathrm{SMEFT}}^{d_i \to u_j \ell \nu} \supset \quad & [C_{V_L}]_{ijkl} (\bar{Q}^i \gamma_\mu \sigma^a Q^j)(\bar{L}^k \gamma^\mu \sigma_a L^l) \\
& + [C_{S_L}]_{ijkl} (\bar{Q}^i u_R^j) i\sigma^2 (\bar{L}^k e_R^l) + [C_{S_R}]_{ijkl} (\bar{Q}_R^i q^j)(\bar{L}^k e_R^l) \\
& + [C_T]_{ijkl} (\bar{Q}^i \sigma_{\mu\nu} u_R^j) i\sigma^2 (\bar{L}^k \sigma^{\mu\nu} e_R^l) + \mathrm{h.c.}.
\end{aligned}
\tag{6}
$$

Here $Q_i = (V_{ji}^* u_L^j, d_L^i)^T$ and $L_i = (U_{ji}^* \nu_L^j, \ell_L^i)^T$ are the LH quark and lepton doublets in the basis aligned with diagonal down-quarks and charged leptons, $U$ is the $3 \times 3$ PMNS mixing matrix for neutrinos and $[C_I]$ with $I \in \{V_L, S_X, T\}$ are the Wilson coefficients. Interestingly, because of $\mathrm{SU}(2)_L$ invariance, these operators, besides giving rise to charged current transitions $d^i \to u^j \ell^k \nu^k$ will also generate neutral current transitions of the form $u^i \bar{u}^j, d^i \bar{d}^j \to \ell^k \ell^l$. We now need to fix the flavor structure in (6). A reasonable assumption is to impose a global flavor symmetry $U(2)_{q_{1,2}} \times U(2)_{\ell_{1,2}}$ acting non-trivially on the first two generations [21]. In the limit this symmetry is exact, one is left only with third generation currents $[C_I]_{ijkl} = \delta_{i3} \delta_{j3} \delta_{k3} \delta_{l3} C_I$. The necessary couplings between different quark generations arise through CKM mixing. Notice that the results we present here should not change much if this global $U(2)^2$ symmetry is slightly broken. Once electro-weak symmetry is spontaneously broken, an immediate consequence of this flavor structure is the appearance of flavor diagonal transitions $t \to b$ and neutral currents $b \to b$, $t \to t$ that are $V_{cb}^{-1}$ enhanced with respect to the $b \to c$ transitions for $R_{D^{(*)}}$. Of particular interest are the potentially large BSM contributions to $b\bar{b}, c\bar{c} \to \tau\bar{\tau}$ scattering [10]. Indeed, since the characteristic scale of NP lies below the TeV scale and the neutral current couplings must be of order $\mathcal{O}(1)$ this opens the possibility for *directly* searching for the NP responsible for $R_{D^{(*)}}$ in $\tau\bar{\tau}$ Drell-Yan production at the LHC. As we show bellow, a close look a the existing LHC data in the $\tau\bar{\tau}$ tails exclude some of the standard proposals solving the $R_{D^{(*)}}$ anomaly, namely, the vector $W'$ boson and the charged scalar $H^+$.

## 3 Simplified dynamical models

In order to perform reliable high-$p_T$ studies at colliders it is necessary to go beyond the SMEFT framework. One first needs to identify all possible tree level mediators that give rise to the effective operators in (6) after integrating them out at the cutoff scale. The new degrees of freedom are then described by a simplified dynamical model, i.e. a minimalistic Lagrangian with a small number of free parameters (couplings, masses and widths) describing the interactions of the mediator with the relevant fermionic currents.

All possible single tree-level mediators contributing to the chiral structure of $R_{D^{(*)}}$ can be classified with their spin and color. The color-neutral states are a scalar doublet $H' \sim (\mathbf{1}, \mathbf{2})_{-\frac{1}{2}}$ and a vector triplet $W'^a \sim (\mathbf{1}, \mathbf{3})_0$ (with the same quantum numbers as the SM EW triplet $W^a$), where we use the notation $(\mathrm{SU}(3)_c, \mathrm{SU}(2)_L)_{\mathrm{U}(1)_Y}$ for the SM group representations. The colourful states are scalar leptoquarks with representations $S_1 \sim (\bar{\mathbf{3}}, \mathbf{1})_{1/3}$, $R_2 \sim (\mathbf{3}, \mathbf{2})_{7/6}$, $\tilde{R}_2 \sim (\mathbf{3}, \mathbf{2})_{1/6}$ and $S_3 \sim (\bar{\mathbf{3}}, \mathbf{3})_{1/3}$, or vector leptoquarks with representations $U_1 \sim (\mathbf{3}, \mathbf{1})_{2/3}$ and $U_3 \sim (\mathbf{3}, \mathbf{3})_{2/3}$. At the LHC, these states will give rise to $\tau\bar{\tau}$ production in two different channels depending on the color contraction. The neutral components of the heavy color singlets will be produced on-shell via $b\bar{b}$ annihilation and decay into $\tau\bar{\tau}$ pairs, as shown in the Feynman diagram of Fig. 2 (left). These heavy states will give rise to a resonant bump in the high-mass region of the di-tau invariant mass spectrum. On the other hand, LQs will give rise to non-resonant $\tau\bar{\tau}$ pair production in the $t$-channel from bottom fusion, as shown in Fig. 2 (right). The effect of this process will be to produce an overall excess of events in the high-

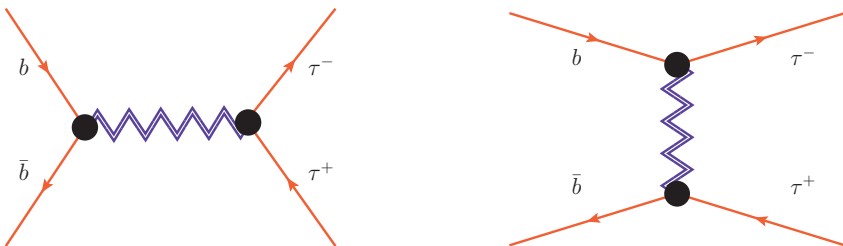

Figure 2: Diagrammatic representations of NP contributions to $b\bar{b} \to \tau^+\tau^-$ at the LHC. (left) $s-$channel resonant exchange of color-neutral mediators, (right) $t-$channel non-resonant exchange of leptoquarks.

mass region of the di-tau invariant mass spectrum.

### 3.1 Color-neutral models

**Vector triplet.** This massive vector decomposes as $W'^a \sim W'^\pm, Z'$ and couples to the SM fermions via

$$\mathcal{L}_{W'} = -\frac{1}{4}W'^{a\mu\nu}W'^a_{\mu\nu} + \frac{M^2_{W'}}{2}W'^{a\mu}W'^a_\mu + W'^a_\mu J^{a\mu}_{W'},$$
$$J^{a\mu}_{W'} \equiv \lambda^q_{ij}\bar{Q}_i\gamma^\mu\sigma^a Q_j + \lambda^\ell_{ij}\bar{L}_i\gamma^\mu\sigma^a L_j. \tag{7}$$

Since the largest effects should involve $B$-mesons and tau leptons we assume $\lambda^{q(\ell)}_{ij} \simeq g_{b(\tau)}\delta_{i3}\delta_{j3}$, consistent with the $U(2)^2$ flavor symmetry [21]. In addition, electroweak precision data requires the masses of $W'$ and $Z'$ to be degenerate up to small corrections of order $\mathcal{O}(1\%)$ [22]. This has two important implications: **(i)** it allows to correlate NP in charged currents at low energies with neutral resonance searches at high-$p_T$; **(ii)** LEP bounds on pair production of charged bosons decaying to $\tau\nu$ final states [23] can be used to constrain the $Z'$ mass from below at $M_{Z'} \simeq M_{W'} \sim 100$ GeV. Integrating out the heavy $W'^a$ at tree level and expanding the SU(2)$_L$ indices give rise to the matching condition for the $V-A$ operator

$$g_{V_L} = -\frac{g_b g_\tau v^2}{M^2_{W'}}. \tag{8}$$

The resolution of the $R_{D^{(*)}}$ anomaly via $W'^\pm$ requires this Wilson coefficient $g_{V_L}$ to be large, leading at the same time to an enhancement in $b\bar{b} \to Z' \to \tau\bar{\tau}$ production at the LHC.

**Scalar doublet.** This massive state decomposes as $H' \sim (H^+, (H^0 + iA^0)/\sqrt{2})$ and has a renormalizable Lagrangian of the form

$$\mathcal{L}_{H'} = |D^\mu H'|^2 - M^2_{H'}|H'|^2 - \lambda_{H'}|H'|^4 - \delta V(H', H)$$
$$- Y_b\bar{Q}_3 H' b_R - Y_c\bar{Q}_3\tilde{H}' c_R - Y_\tau\bar{L}_3 H'\tau_R + \text{h.c.}, \tag{9}$$

where $\tilde{H}' = i\sigma^2 H'^*$ and $\delta V(H', H)$ parametrizes additional terms in the scalar potential which split the masses of $A^0$, $H^0$, $H^+$ and mix $H^0$ with the SM Higgs boson away from the alignment (inert) limit. The corresponding high-$p_T$ signatures at the LHC are given by $b\bar{b} \to (H^0, A^0) \to \tau^+\tau^-$. On the other hand, the $b \to c$ transition for $R_{D^{(*)}}$ is mediated by the charged component $H^\pm$. Integrating out this state gives rise to the scalar operators and $\mathcal{O}_{S_L}$ and $\mathcal{O}_{S_R}$. As pointed out in [14] the current bound on the $B_c$-lifetime is strong enough to

Table 1: Summary of the LQ models which can accommodate $R_{K^{(*)}}$ (first column), $R_{D^{(*)}}$ (second column), and both $R_{K^{(*)}}$ and $R_{D^{(*)}}$ (third column) without introducing phenomenological problems. See Ref. [11] for details.

| Model | $R_{K^{(*)}}$ | $R_{D^{(*)}}$ | $R_{K^{(*)}}$ & $R_{D^{(*)}}$ |
|---|---|---|---|
| $S_1 \sim (\bar{\mathbf{3}}, \mathbf{1})_{1/3}$ | ✗ | ✓ | ✗ |
| $R_2 \sim (\mathbf{3}, \mathbf{2})_{7/6}$ | ✗ | ✓ | ✗ |
| $\widetilde{R_2} \sim (\mathbf{3}, \mathbf{2})_{1/6}$ | ✗ | ✗ | ✗ |
| $S_3 \sim (\bar{\mathbf{3}}, \mathbf{3})_{1/3}$ | ✓ | ✗ | ✗ |
| $U_1^\mu \sim (\mathbf{3}, \mathbf{1})_{2/3}$ | ✓ | ✓ | ✓ |
| $U_3^\mu \sim (\mathbf{3}, \mathbf{3})_{2/3}$ | ✓ | ✗ | ✗ |

exclude the parameter space necessary to explain $R_{D^{(*)}}$. For this reason we do not discuss this model any further[1].

## 3.2 Leptoquark models

LQs have recently gained attention as possible solutions to the $B$-anomalies. Out of the 12 possible LQ states [24] respecting the SM gauge symmetry, only a few can explain the anomalies. The current status of these models is described in Table 1, see Ref. [11] for more details. There are three minimal scenarios that solve the $B$-anomalies: (i) one vector $U_1$ [25] or two pairs of scalars (ii) $R_2$ and $S_3$ [12] and (iii) $S_1$ and $S_3$ [25, 26]. The relevant simplified models for each of these scenarios are described below (we do not include here the simplified model for $S_3$ LQ since it has a small impact on LHC phenomenology). On a side note, UV completions for these three LQ scenarios have been proposed in the literature. For example, in Refs. [27, 28] $U_1$ is the Pati-Salam gauge boson of the SU(4) gauge group. In [12, 29], $R_2$ and $S_3$ come from an SU(5) GUT framework, while in [30] $S_1$ and $S_3$ are taken as pseudo Nambu-Goldstone modes from a strongly coupled theory.

**Vector singlet $U_1$.** First we consider the vector LQ $U_1$, which received considerable attention because it can provide a simultaneous explanation to the anomalies in $b \rightarrow s$ and $b \rightarrow c$ transitions [25]. The most general Lagrangian consistent with the SM gauge symmetry allows couplings to both LH and RH fermions, namely,

$$\mathcal{L}_{U_1} = x_L^{ij} \bar{Q}_i \gamma_\mu U_1^\mu L_j + x_R^{ij} \bar{d}_{Ri} \gamma_\mu U_1^\mu \ell_{Rj} + \text{h.c.}, \tag{10}$$

where $x_L^{ij}$ and $x_R^{ij}$ are the couplings. Furthermore, this scenario also contributes to $b \rightarrow c \ell \bar{\nu}_{\ell'}$ by giving rise to the effective coefficient

$$g_{V_L} = \frac{v^2}{2m_{U_1}^2} (x_L^{b\ell})^* \left[ x_L^{b\ell'} + \frac{V_{cs}}{V_{cb}} x_L^{s\ell'} + \frac{V_{cd}}{V_{cb}} x_L^{d\ell'} \right], \tag{11}$$

where the second and third terms in $g_{V_L}$ vanish in the limit where the $U(2)^2$ flavor symmetry is exact. Relaxing this criteria by explicitly breaking the flavor symmetry with small $x_L^{b\mu}, x_L^{s\mu}, x_L^{s\tau} \neq 0$, gives rise to (suppressed) interactions in the second generations necessary for $b \rightarrow s\mu\mu$ transitions but also generates additional sources for $b \rightarrow c\tau\nu$. As a consequence,

---

[1]In any case, direct searches for the neutral scalars $H^0/A^0$ in di-tau tails with current LHC data also exclude this scenario [10].

a mildly broken $U(2)^2$ flavor symmetry can then explain both $R_{D^{(*)}}$ and $R_{K^{(*)}}$ with one single $V-A$ operator.

**Scalar singlet $S_1$.** The most general Yukawa Lagrangian for $S_1$ reads

$$
\begin{aligned}
\mathcal{L}_{S_1} &= y_L^{ij}\,\overline{Q^C}i\tau_2 L_j\, S_1 + y_R^{ij}\,\overline{u_{Ri}^C}e_{Rj}\,S_1 + \text{h.c.} \\
&= S_1\Big[\big(V^* y_L\big)_{ij}\,\overline{u_{Li}^C}\ell_{Lj} - y_L^{ij}\,\overline{d_{Li}^C}\nu_{Lj} + y_R^{ij}\,\overline{u_{Ri}^C}\ell_{Rj}\Big] + \text{h.c.},
\end{aligned}
\tag{12}
$$

where $y_L$ and $y_R$ are general $3 \times 3$ Yukawa matrices. Here we omitted the terms involving diquark couplings which must be forbidden to guarantee the stability of the proton. Once integrating out this LQ state at tree level at the matching scale $\mu = m_{S_1}$, we find $V-A$ and Scalar/Tensor contributions to $b \to c\ell\,\bar{\nu}_{\ell'}$:

$$
g_{V_L} = \frac{v^2}{4V_{cb}}\frac{y_L^{b\ell'}\big(V y_L^*\big)_{c\ell}}{m_{S_1}^2},
\tag{13}
$$

$$
g_{S_L} = -4\,g_T = -\frac{v^2}{4V_{cb}}\frac{y_L^{b\ell'}\big(y_R^{c\ell}\big)^*}{m_{S_1}^2}.
\tag{14}
$$

The $V-A$ operator can fully accommodate $R_{D^{(*)}}$ on its own. Another option is to fit the anomaly with the Scalar/Tensor combination for real Wilson coefficients satisfying $g_{S_L} = -4g_T$ at the cutoff scale (see Fig. 1 (right)).

**Scalar doublet $R_2$.** The most general Lagrangian describing the Yukawa interactions of $R_2$ can be written as

$$
\mathcal{L}_{R_2} = y_R^{ij}\,\overline{Q}_i\ell_{Rj}R_2 - y_L^{ij}\,\overline{u}_{Ri}R_2 i\tau_2 L_j + \text{h.c.},
\tag{15}
$$

where $y_L$ and $y_R$ are the Yukawa matrices, and $\text{SU}(2)_L$ indices have been omitted for simplicity. More explicitly, in terms of the electric charge eigenstates $R_2^{(Q)}$, the Lagrangian (15) can be decomposed as

$$
\begin{aligned}
\mathcal{L}_{R_2} &= (V y_R)_{ij}\,\overline{u}_{Li}\ell_{Rj}R_2^{(5/3)} + (y_R)_{ij}\,\overline{d}_{Li}\ell_{Rj}R_2^{(2/3)} \\
&\quad + (y_L)_{ij}\bar{u}_{Ri}\nu_{Lj}R_2^{(2/3)} - (y_L)_{ij}\bar{u}_{Ri}\ell_{Lj}R_2^{(5/3)} + \text{h.c.}.
\end{aligned}
\tag{16}
$$

Furthermore, this $R_2$ contributes to the transition $b \to c\tau\bar{\nu}_{\ell'}$ purely via the Scalar/Tensor solution. The tree-level matching at the scale $\mu = m_{R_2}$ is given by the Wilson coefficients:

$$
g_{S_L} = 4\,g_T = \frac{v^2}{4V_{cb}}\frac{y_L^{c\ell'}\big(y_R^{b\ell}\big)^*}{m_{R_2}^2}.
\tag{17}
$$

This scenario can accommodate the observed experimental deviations in $R_{D^{(*)}}$ for purely imaginary couplings, as can be seen in Fig. 1, and in Refs. [31–33].

# 4 LHC phenomenology

## 4.1 High-mass di-tau tails

We now confront the simplified models with $pp \to Z' \to \tau\bar{\tau}$ resonance searches at the LHC. Constraints were first derived in [10] using both 8 TeV and 13 TeV ATLAS searches in the

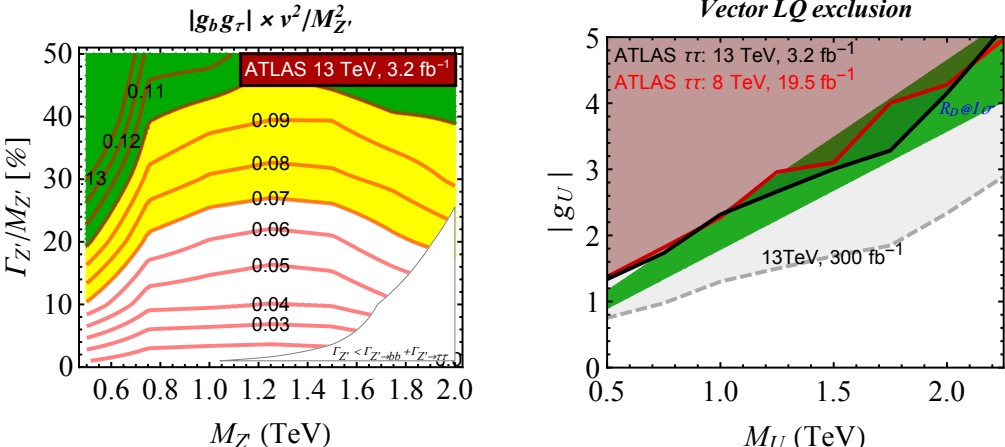

Figure 3: (left plot) 13 TeV exclusion limits on the $b\bar{b} \to Z' \to \tau\tau$ resonance at 3.2 fb$^{-1}$. Isolines shown in red represent upper limits on the combination $g_{V_L} = |g_b g_\tau| \times v^2/M_{Z'}^2$ as a function of the $Z'$ mass and total width. The $R_{D^{(*)}}$ preferred regions at 68% and 95% CL are shaded in green and yellow, respectively. (right plot) 8 TeV (13 TeV) ATLAS $\tau\bar{\tau}$ search exclusion limits are shown in red (black) and $R_{D^{(*)}}$ preferred region in green for the vector LQ model. Here the coupling is defined by $g_U \equiv x_L^{b\tau}$. Projected 13 TeV limits for 300 fb$^{-1}$ are shown in gray.

hadronic tau category at 20 fb$^{-1}$ [34] and 3.2 fb$^{-1}$ [35], respectively. One important result from this study is that color-neutral NP models for $R_{D^{(*)}}$ with perturbative couplings, i.e. $W'$ and $H'$, are excluded by $\tau\bar{\tau}$ data. Results for the vector triplet are given in Fig. 3 (left), where we show the 95% CL upper limits for fixed values of $g_{V_L} = |g_b g_\tau| v^2/M_{Z'}^2$ as red iso-contours in the mass versus width plane of the $Z'$ boson. The allowed region in green (yellow) accommodates the $R_{D^{(*)}}$ anomaly at $1\sigma$ ($2\sigma$). This shows that the width of the $Z'$ boson must be unnaturally broad, surpassing 30-40%, in order to evade these direct search limits. Similar conclusions can be reached for the neutral scalar and pseudoscalar $H^0/A^0$.

For LQ models the limits from recasting di-tau resonance searches are evidently much weaker. For example, for the vector $U_1$ LQ the 95% CL exclusion limits in the coupling[2] vs mass plane shown in Fig. 3 (right) is given by the gray region (red region) for the 13 TeV (8 TeV) LHC searches. Notice that the LHC is starting to probe the green band that explains the $B$-anomaly at $1\sigma$ for LQ couplings in the limit of exact $U(2)^2$ flavor symmetry. A naive projection of these limits to a higher luminosity of 300 fb$^{-1}$ shows that the LHC will completely probe this scenario. Nonetheless, as shown in [25], explicitly breaking the global $U(2)^2$ symmetry by allowing for a (small) non-zero $x_L^{s\tau}$ coupling in (11) leads to a reduction of $x_L^{b\tau}$ in the $R_{D^{(*)}}$ fit. This particular scenario evades these di-tau bounds and gives motivation for future HL-LHC studies. In fact, a more recent $\tau\bar{\tau}$ search by ATLAS [36] at a luminosity of 36.1 fb$^{-1}$ was used in [11] to update the 3.2 fb$^{-1}$ limits on both vector and scalar LQs. In Fig. 4 we provide the 95% CL exclusion limits in the coupling $y^{q\ell}$ ($x^{q\ell}$) vs mass plane for several scalar (vector) LQs and different initial sea quarks, $b\bar{b} \to \tau\bar{\tau}$ (solid blue), $c\bar{c} \to \tau\bar{\tau}$ (solid green) and $s\bar{s} \to \tau\bar{\tau}$ (solid red). Similarly, we have also included in dashed lines and the same color code, the exclusion limits from recasting a $Z'$ resonance search by ATLAS [37] in di-muon tails at 36.1 fb$^{-1}$ for $t$-channel LQ exchange in $pp \to \mu\bar{\mu}$. While these bounds are not relevant for the $R_{D^{(*)}}$ anomaly, the di-muon tails can be constraining for certain NP models for $R_{K^{(*)}}$ at tree-level [38] and very constraining for one-loop models for $R_{K^{(*)}}$ [39].

---

[2]In the plot we have redefined the coupling to $g_U \equiv x_L^{b\tau}$.

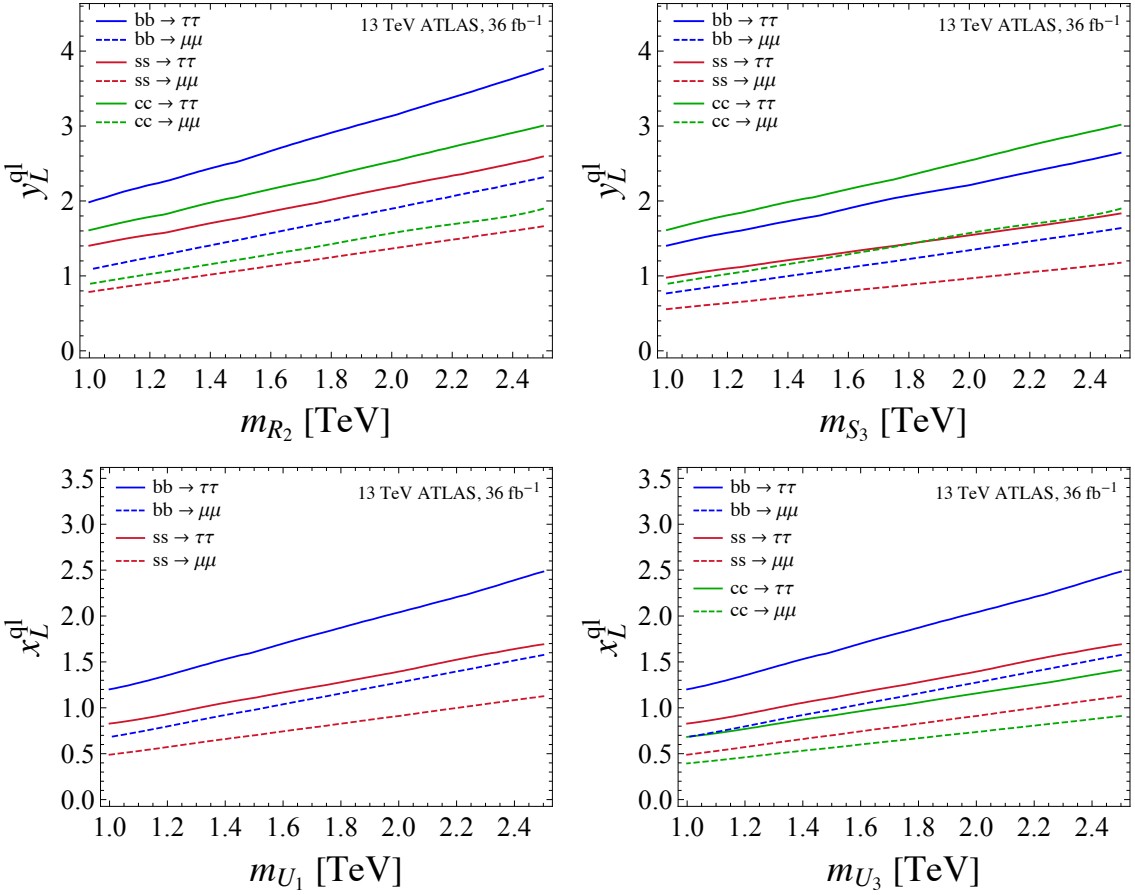

Figure 4: The top panel (lower panel) shows current limits in the coupling vs mass plane for several scalar LQ (vector LQ) models from LHC searches in $pp \to \tau\bar{\tau}, \mu\bar{\mu}$ high-$p_T$ tails at 13 TeV with 36 fb$^{-1}$ of data. The solid and dashed lines represent limits from di-tau and di-muon searches, respectively, for different initial quarks while turning one scalar (vector) LQ coupling $y_L^{ql}$ ($x_L^{ql}$) at a time.

## 4.2 Leptoquark searches

The most relevant LQ process at the LHC is pair production $gg\,(q\bar{q}) \to$ LQ$^\dagger$LQ. ATLAS and CMS have searched for this process in different decay channels into second and/or third generation quarks and leptons, LQ$^\dagger$LQ $\to q\bar{q}\ell\bar{\ell}, q\bar{q}\nu\bar{\nu}$. As a result, these searches lead to useful model independent bounds on both the mass and branching fractions of the LQ. For example, the vector $U_1$ LQ in the exact $U(2)^2$ flavor limit, when produced in pairs decays into $c\bar{c}\nu\nu$ and $b\bar{b}\tau\tau$ each with a 25% branching ratio. The best limit on the mass is currently at $M_U > 1.5$ TeV from a LQ search by CMS for final state dijets plus missing energy. In Table 2 we list the most recent lower limits on the masses of second/third generation scalar and vector LQs relevant for the $B$-anomalies, for benchmark branching ratios set to $\beta = 1\,(0.5)$. For more details on LQ pair searches see [11, 27].

Besides pair production and $t$-channel Drell-Yan production, LQs can also be singly produced at the LHC via $qg \to$ LQ$\ell$, LQ$\nu$. This mode is usually sub-dominant for small LQ couplings to fermions and only becomes important when the couplings are large enough, usually for LQ couplings above $y^{q\ell}, x^{q\ell} \sim 2$. Interestingly, given that non-resonant Drell-Yan LQ production, LQ pair production and single LQ production all scale differently with powers of the

Table 2: Summary of the current limits from LQ pair production searches at the LHC. In the first column we give the searched final states and in the second column the LQs for which this search is relevant. In the next two columns we present the current limits on the mass for scalar and vector LQs, respectively, for $\beta = 1$ ($\beta = 0.5$). In the last column we display the value of the LHC luminosity for each search along with the experimental references. Note that "$j$" denotes any jet originating from a charm or a strange quark.

| Decays | LQs | Scalar LQ limits | Vector LQ limits | $\mathcal{L}_{\text{int}}$ / Ref. |
|---|---|---|---|---|
| $jj\,\tau\bar{\tau}$ | $S_1, R_2, S_3, U_1, U_3$ | – | – | – |
| $b\bar{b}\,\tau\bar{\tau}$ | $R_2, S_3, U_1, U_3$ | 850 (550) GeV | 1550 (1290) GeV | 12.9 fb$^{-1}$ [40] |
| $t\bar{t}\,\tau\bar{\tau}$ | $S_1, R_2, S_3, U_3$ | 900 (560) GeV | 1440 (1220) GeV | 35.9 fb$^{-1}$ [41] |
| $jj\,\mu\bar{\mu}$ | $S_1, R_2, S_3, U_1, U_3$ | 1530 (1275) GeV | 2110 (1860) GeV | 35.9 fb$^{-1}$ [42] |
| $b\bar{b}\,\mu\bar{\mu}$ | $R_2, U_1, U_3$ | 1400 (1160) GeV | 1900 (1700) GeV | 36.1 fb$^{-1}$ [27] |
| $t\bar{t}\,\mu\bar{\mu}$ | $S_1, R_2, S_3, U_3$ | 1420 (950) GeV | 1780 (1560) GeV | 36.1 fb$^{-1}$ [39,43] |
| $jj\,\nu\bar{\nu}$ | $R_2, S_3, U_1, U_3$ | 980 (640) GeV | 1790 (1500) GeV | 35.9 fb$^{-1}$ [44] |
| $b\bar{b}\,\nu\bar{\nu}$ | $S_1, R_2, S_3, U_3$ | 1100 (800) GeV | 1810 (1540) GeV | 35.9 fb$^{-1}$ [44] |
| $t\bar{t}\,\nu\bar{\nu}$ | $R_2, S_3, U_1, U_3$ | 1020 (820) GeV | 1780 (1530) GeV | 35.9 fb$^{-1}$ [44] |

LQ coupling, implies that all three modes give complementary bounds in parameter space, as illustrated in Fig. 5 in Ref. [45]. For this reason all modes should be thoroughly searched by both experimental collaborations at the LHC.

## 5 Closing the window on LQ solutions

In this section we show in more detail how direct searches at the LHC with current and future data are starting to probe the interesting regions of parameter space for the LQ solutions to the *B*-anomalies. We illustrate this with the vector $U_1$ and the scalar $R_2$. As for the $S_1$ scalar LQ, the reader is referred to Ref. [25, 30] for the LHC analysis. We also highlight the complementarity between di-tau searches and rare LFV searches in *B* and $\tau$ decays. As shown below, this complementarity can be jointly exploited in the near future by the HL-LHC and low-energy experiments for ultimately testing these model.

### 5.1 The vector $U_1$ LQ

This LQ model equipped with a minimally broken $U(2)^2$ global symmetry is the only single particle[3] able to generate $R_{K^{(*)}} < R_{K^{(*)}}^{\text{SM}}$ and $R_{D^{(*)}} > R_{D^{(*)}}^{\text{SM}}$ while respecting all low energy flavor and LHC constraints. To explore this in more detail, we assume for the $U_1$ Lagrangian in (10) the following generic structure for the Yukawa matrices:

$$x_L = \begin{pmatrix} 0 & 0 & 0 \\ 0 & x_L^{s\mu} & x_L^{s\tau} \\ 0 & x_L^{b\mu} & x_L^{b\tau} \end{pmatrix}, \qquad x_R = 0. \tag{18}$$

Because of the stringent limits from $\mu - e$ conversion on nuclei, atomic parity violation and

---

[3]Declared *Particle of the Year* at CKM 2018.

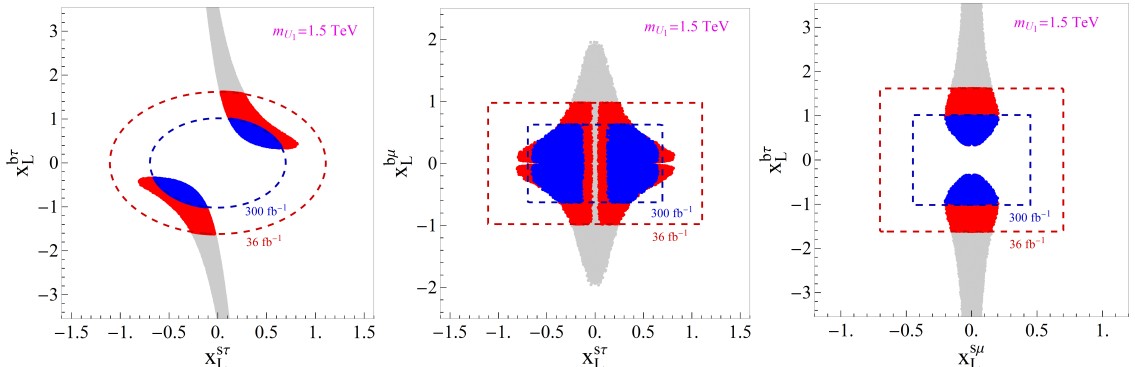

Figure 5: Scatter plots showing the allowed regions of parameter space for different combination of $U_1$ LQ couplings assuming $m_{U_1} = 1.5$ TeV. For the color code go to text.

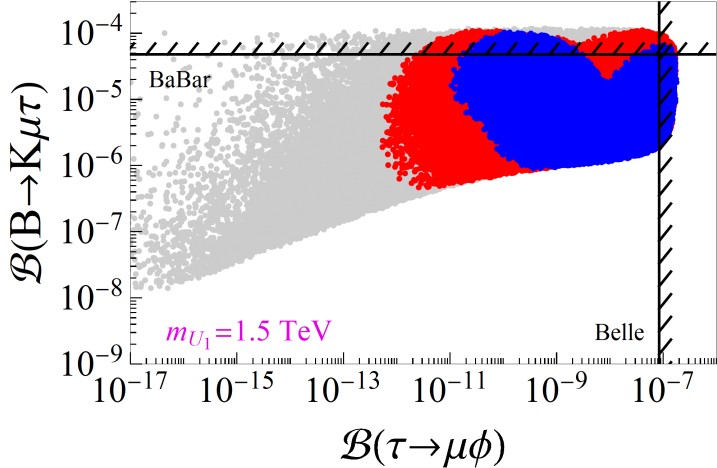

Figure 6: Br($B \rightarrow K\mu\tau$) is plotted against $\mathcal{B}(\tau \rightarrow \mu\phi)$ for the $U_1$ model. Color code is the same as in Fig. 5. Current bounds on these two decays, as respectively established by BaBar [46] and by Belle [47], are also shown.

$\mathcal{B}(K \rightarrow \pi\nu\bar{\nu})$, the coupling to the first generation are set to zero. We performed a fit to the $B$-anomalies, $K \rightarrow \mu\bar{\nu}$, $D_{(s)} \rightarrow \tau\bar{\nu}$ and $B \rightarrow \tau\bar{\nu}$, the ratio $R_D^{\mu/e} = \text{Br}(B \rightarrow D\mu\bar{\nu})/\text{Br}(B \rightarrow De\bar{\nu})$ as well as the LFV processes $B \rightarrow K\mu\tau$ and $\tau \rightarrow \mu\phi$. In the fit we have left out one-loop observables [48]. We also fixed the benchamrk mass at $m_{U_1} = 1.5$ TeV, which is the lowest $U_1$ mass not yet excluded by vector LQ pair production searches at the LHC [44]. Results for the available parameter space are given by the scatter plots in Fig. 5. The selected points correspond to those which fall within a $2\sigma$ range from the best fit point. These points are then compared with the limits deduced from the direct LHC searches in $pp \rightarrow \tau\bar{\tau}$, $\mu\bar{\mu}$ tails[4]. The points excluded by direct searches based on a current (projected) LHC luminosity of 36 fb$^{-1}$ (300 fb$^{-1}$) are shown in gray (red) while the blue points are those that would survive for a projected luminosity at 300 fb$^{-1}$.

An important observation from Fig. 5 is that in order to avoid the current $\tau\bar{\tau}$ bounds from LHC a non-zero (small) value for $|x_L^{s\tau}|$ is necessary. In addition to this, the requirement of

---

[4]We also include contributions from (sub-leading) Cabibbo suppressed contributions $u\bar{u} \rightarrow \ell\bar{\ell}$.

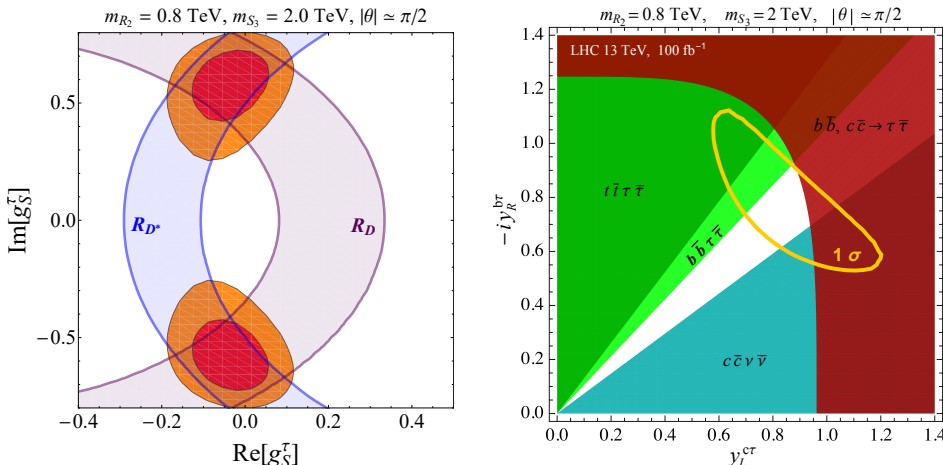

Figure 7: (Left) Results of the flavor fit in the $g_{S_L}$ plane. The allowed $1\sigma$ ($2\sigma$) regions are rendered in red (orange). Separate constraints from $R_D$ and $R_{D^*}$ at $2\sigma$ accuracy are shown by the blue and purple regions, respectively. (Right) Summary of the LHC exclusion limits (colored regions) for each LQ process at a projected luminosity of $100\,\text{fb}^{-1}$ for $m_{R_2} = 800\,\text{GeV}$, $m_{S_3} = 2\,\text{TeV}$, and $|\theta| \approx \pi/2$. The region inside the yellow contour corresponds to the $1\sigma$ fit to the low-energy observables. See text for detail.

non-vanishing $x_L^{s\mu}$ and $x_L^{b\mu}$ to explain the $R_{K^{(*)}}$ anomaly have an important impact on the LFV decays in this model, in particular the modes $B \to K\mu\tau$ and $\tau \to \mu\phi$. This has been illustrated in Fig. 6. Notice that the projected LHC bounds lead to a lower bound for each of these modes of order $\mathcal{O}(10^{-7})$ for Br($B \to K\mu\tau$) and $\mathcal{O}(10^{-11})$ for Br($\tau \to \mu\phi$). We have also included the current bounds from BaBar [46] and by Belle [47] for each LFV mode (solid hashed lines). We see that lowering the upper bound on Br($B \to K\mu\tau$) and Br($\tau \to \mu\phi$) at the LHCb and/or Belle II can have a major impact on the model building by further restraining the parameter space. Here, for definiteness we focus on $B \to K\mu\tau$, but the discussion would be completely equivalent if we used $B_s \to \mu\tau$ or $B \to K^*\mu\tau$, because their branching fractions are known to be related.

## 5.2 The GUT inspired scalar LQs

A simultaneous solution to the $B$-anomalies can arise from a UV complete model based on SU(5) Grand Unified Thoery (GUT), where two light scalar LQs, $R_2$ and $S_3$, appear at the TeV scale [12]. Once integrated out, the doublet gives rise to the Scalar/Tensor operators explaining $R_{D^{(*)}}$ while the triplet generates the $V-A$ operator necessary for explaining $R_{K^{(*)}}$. Furthermore, $S_3$ and $R_2$ have a common origin in the UV given that both states are (partially) embedded in the same scalar representation **45** of SU(5). As a consequence, at low energies both LQs share the same Yukawa matrix, $y_L$, up to a sign. We now focus on $R_2$ since it drives the phenomenology at the LHC. Once rotating Eq. (15) to the mass eigenbasis, the simplest flavor texture for this model is

$$y_R = \begin{pmatrix} 0 & 0 & 0 \\ 0 & 0 & 0 \\ 0 & 0 & y_R^{b\tau} \end{pmatrix}, \quad y_L = \begin{pmatrix} 0 & 0 & 0 \\ 0 & c_\theta & -s_\theta \\ 0 & s_\theta & c_\theta \end{pmatrix} \begin{pmatrix} 0 & 0 & 0 \\ 0 & y_L^{c\mu} & y_L^{c\tau} \\ 0 & 0 & 0 \end{pmatrix}, \quad (19)$$

where $\theta$ is a mixing angle, $s_\chi \equiv \sin\chi$, $c_\chi \equiv \cos\chi$, $y_R^{b\tau}$ is a complex Yukawa coupling and both $y_L^{c\mu}$ are $y_L^{b\tau}$ are real Yukawa couplings. These 4 parameters along with the two LQ

masses are the total six parameters of this model. As a benchmark for the LHC analysis we set $m_2 = 800$ GeV (and $m_{S_3} = 2$ TeV). In Fig. 7 (left) we show in red (orange) the $1\sigma$ ($2\sigma$) region from a global fit to all relevant low energy observables in the complex plane of the scalar Wilson coefficient $g_{S_L} = 4g_T$. We also included in the same figure the $2\sigma$ regions accommodating $R_D$ in purple and $R_{D^*}$ in blue. The low energy fit requires the mixing angle to satisfy $\sin^2(2\theta) \approx 0$, leading to two possible solutions $\theta \approx \{0, \pi/2\}$. Interestingly, the current limit for the LFV observable $\mathrm{Br}(\tau \to \mu\phi)$ breaks this degeneracy and fixes the mixing angle to be near maximal, i.e. $\theta \approx \pi/2$.

To finilize, we now confront this model to the direct searches at the LHC. The main contributions to di-tau production come from the $t$-channel exchange of the components $R_2^{(5/3)}$ and $R_2^{(2/3)}$ in $c\bar{c} \to \tau\bar{\tau}$ and $b\bar{b} \to \tau\bar{\tau}$, respectively[5]. Our results for the 95% CL limits in the $y_L^{c\tau}$–($y_R^{b\tau}/i$) plane are given by the red exclusion region in Fig. 7 (right) at a projected LHC luminosity of $100\,\mathrm{fb}^{-1}$ for the benchmark masses and $|\theta| \approx \pi/2$. Since the $R_2$ LQ needs to be quite light, we also took into account bounds from pair production. To set these limits we used the CMS search [40] targeting $pp \to (R_2^{(2/3)})^* R_2^{(2/3)}$ decaying into $b\bar{b}\tau\bar{\tau}$ final states and the multi-jet plus missing energy search [44] for decays into $c\bar{c}\nu\bar{\nu}$ final states. The 95% CL exclusion limits are shown by the light green and turquoise regions in Fig. 7 for a luminosity of $100\,\mathrm{fb}^{-1}$. As for pair produced $R_2^{(5/3)}$ states decaying into $t\bar{t}\tau\bar{\tau}$ we employed Ref. [41]. This result corresponds to the dark green exclusion region in Fig. 7 (right). The $1\sigma$ region satisfying all low-energy data, including the $B$-anomalies, is given in the same figure by the thick yellow contour. Interestingly, for such a low $R_2$ mass there is still allowed parameter space, which could eventually be covered by the HL-LHC. Notice that for a slightly higher masses of $\approx 1$ TeV the LHC pair production bounds relax considerably. In this model we have also found a complementarity between high energy and low energy observables. In particular, between high-mass di-tau tails, the rare decay $\mathrm{Br}(B \to K\nu\bar{\nu})$ and the LFV decay mode $\mathrm{Br}(B \to K\mu\tau)$, for more details see [12]. Experimental inputs from HL-LHC, Belle II and LHCb could ultimately test the predictions for these observables in the near future. Another interesting prediction, which is a consequence of the almost purely imaginary coupling $y_R^{b\tau}$, is a new source of CP violation. See for instance [49] for a recent analysis of the current and projected limits from electric dipole moment searches.

## 6 Conclusion

In this proceedings we have discussed the impact of direct searches at the LHC on NP models for the $B$-anomalies. Using model independent arguments we correlated semi-tauonic $B$-decays with $pp \to \tau\bar{\tau}$ production at the LHC and showed how current data from the $\tau\bar{\tau}$ invariant mass tails rules out color-neutral mediators, $W'$ and $H^+$, as solutions to the $R_{D^{(*)}}$ deviation, leaving LQ models as the most promising NP explanation of the $B$-anomalies. After laying out the full bestiary of LQs, we identified three interesting scenarios: (i) the vector LQ, $U_1$, which happens to be the only single mediator solving simultaneously $R_{D^{(*)}}$ and $R_{K^{(*)}}$, (ii) the scalar doublet $R_2$ for $R_{D^{(*)}}$ combined with a scalar triplet $S_3$ for $R_{K^{(*)}}$ and (iii) the scalar singlet $S_1$ for $R_{D^{(*)}}$ combined with a triplet $S_3$ for $R_{K^{(*)}}$. While these models are not yet excluded by di-tau searches, the relevant portion of parameter space for are currently starting to be probed in direct searches at the LHC in di-tau tails and LQ pair production. A deeper analysis of the vector LQ $U_1$ reveals an interesting complementarity between di-tau tails and low energy LFV decay modes $B \to K\mu\tau$ and $\tau \to \mu\phi$. If with more data, the anomalies persist near their

---

[5]Sub-leading contributions from the more massive $S_3$ to $b\bar{b} \to \tau\bar{\tau}$ have also been included in this analysis. These effects are negligible at the LHC.

current central values, then improving the LFV decay bounds between one and two orders of magnitude at Belle II or LHCb, can either exclude or, if observed, validate the $U_1$ scenario. Similar conclusions apply to the GUT-inspired scalar LQ solution. In this case the low-energy fit demands a rather light $R_2$ close to 1 TeV, meaning that LQ pair production searches provide an important experimental handle complementary to the di-tau tails. This scalar LQ model should be completely accessible at the HL-LHC in these two channels, as well as low energy decay modes such as $B \to K \nu \bar{\nu}$, $B \to K \mu \tau$ and $\tau \to \mu \phi$ at Belle II and LHCb.

## 7  Acknowledgements

I would like to thank A. Angelescu, D. Bečirević, J. E. Camargo-Molina, A. Celis, I. Doršner, S. Fajfer, A. Greljo, J. F. Kamenik, N. Košnik, O. Sumensari and L. Valle Silva for many illuminating discussions about this intriguing subject.

**Funding information**  The author is supported by the *Young Researchers Programme* of the Slovenian Research Agency under the grant N° 37468.

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
