# Peer review of "LHC searches motivated by recent $B$-anomalies"

_SciPost Physics Proceedings, doi:SciPost Phys. Proc. 1, 021 (2019)_

## Round 1 · Referee Report · Anonymous · 2018-12-6

Strengths
The manuscript is very nice and well written.
Motivated by the tantalizing discrepancies with the SM observed in B mesons decays, it verifies which models candidate to explain the B-anomalies are also compatible with the result of the searches for high mass particles at the LHC.
It finds, for that only few LQ models pass the test of the data and discuss perspective for end of LHC Run 2, Run 3 and HL-LHC.
Weaknesses
Nothing significant
Report
The manuscript discusses the models that can possibly explain the discrepancies with the SM observed in B mesons decays.
The author analyze the implication of the results of the LHC searches for high mass resonances on those models.
In particular the search for heavy neutral particles decaying in two taus final state is analyzed and the results of the ATLAS search for Z' ->tautau final state is reinterpreted in terms of limits on LQ particles which would contribute to the tautau final state via t-channel exchange of a LQ (non-resonant final state).
In addition, exploiting complementary constraints coming from LHC heavy mass particle searches and searches for LFV decays of the B and tau particles, it would be possible to limit further the LQ parameter space in the near future.
Requested changes
1- Small typographic changes requested to the author.

---

## Round 2 · List of Changes

minor typos fixed

---

## Editorial Decision

published